# Relevance of Abnormal KCNN1 Expression and Osmotic Hypersensitivity in Ewing Sarcoma

**DOI:** 10.3390/cancers14194819

**Published:** 2022-10-01

**Authors:** Sebastian Fuest, Christoph Post, Sebastian T. Balbach, Susanne Jabar, Ilka Neumann, Sandra Schimmelpfennig, Sarah Sargin, Elke Nass, Thomas Budde, Sareetha Kailayangiri, Bianca Altvater, Andreas Ranft, Wolfgang Hartmann, Uta Dirksen, Claudia Rössig, Albrecht Schwab, Zoltán Pethő

**Affiliations:** 1Institute of Physiology II, University Münster, 48149 Münster, Germany; 2Department of Pediatric Hematology and Oncology, University Children’s Hospital Münster, 48149 Münster, Germany; 3Pediatrics III, University Hospital Essen, 45147 Essen, Germany; 4Institute of Physiology I, University Münster, 48149 Münster, Germany; 5Division of Translational Pathology, Gerhard-Domagk-Institute of Pathology, University Münster, 48149 Münster, Germany

**Keywords:** K_Ca_2.1 channel, Ewing sarcoma, GGAA microsatellite, regulatory volume decrease

## Abstract

**Simple Summary:**

The main oncogene in Ewing sarcoma directly drives a high expression of a previously unknown variant *KCNN1* (encoding the K_Ca_2.1 channel) that we also verified in samples from >200 patients. Yet, we found that the channel is not functional and does not modulate Ewing sarcoma cell behavior. We could explain this lack of functional impact by the surprising absence of any K_Ca_2.1-carried K^+^ current in Ewing sarcoma cells. However, we show in a proof-of-principle study that the essential lack of a K^+^ conductance can be exploited by applying hypoosmotic stress and effectively and selectively killing the Ewing sarcoma cells.

**Abstract:**

Ewing sarcoma (EwS) is a rare and highly malignant bone tumor occurring mainly in childhood and adolescence. Physiologically, the bone is a central hub for Ca^2+^ homeostasis, which is severely disturbed by osteolytic processes in EwS. Therefore, we aimed to investigate how ion transport proteins involved in Ca^2+^ homeostasis affect EwS pathophysiology. We characterized the expression of 22 candidate genes of Ca^2+^-permeable or Ca^2+^-regulated ion channels in three EwS cell lines and found the Ca^2+^-activated K^+^ channel K_Ca_2.1 (*KCNN1*) to be exceptionally highly expressed. We revealed that *KCNN1* expression is directly regulated by the disease-driving oncoprotein EWSR1-FL1. Due to its consistent overexpression in EwS, *KCNN1* mRNA could be a prognostic marker in EwS. In a large cohort of EwS patients, however, *KCNN1* mRNA quantity does not correlate with clinical parameters. Several functional studies including patch clamp electrophysiology revealed no evidence for K_Ca_2.1 function in EwS cells. Thus, elevated *KCNN1* expression is not translated to K_Ca_2.1 channel activity in EwS cells. However, we found that the low K^+^ conductance of EwS cells renders them susceptible to hypoosmotic solutions. The absence of a relevant K^+^ conductance in EwS thereby provides an opportunity for hypoosmotic therapy that can be exploited during tumor surgery.

## 1. Introduction

Ewing sarcoma (EwS) is the second most common bone malignancy in children and adolescents, driven by a disease-defining translocation between chromosomes 11 and 22 (t(11;22)(q24;q12) [1]. In EwS, this translocation results in a chimeric fusion protein, mainly involving the EWSR1 (Ewing sarcoma breakpoint region 1 gene) and FLI1 (Friend leukemia integration 1 transcription factor), which can bind to GGAA microsatellites as well as to transcriptional complexes and induce expression of numerous target genes [2,3]. Standard combined modality therapy including multiagent chemotherapy, surgical resection and often radiotherapy cures only a proportion of patients, and patients with primary disseminated disease have a dismal prognosis [4].

Bone is a central hub for body mineral and pH homeostasis [5]. As EwS predominantly degrades bone tissue, the cancer cells are surrounded by a unique microenvironment [6]. A main constituent of this microenvironment is Ca^2+^, derived from calcium phosphate in the inorganic bone matrix. Consequently, it is expected that tumor cells have to adapt to an increased extracellular Ca^2+^ ([Ca^2+^]_e_) concentration. Moreover, as Ca^2+^ as well as inflammatory and cellular degradation products are osmotically active, cells have likely to cope with a hyperosmotic milieu. Whether the particularly Ca^2+^-rich microenvironment is relevant to EWS tumorigenesis and progression, how EwS cells regulate their intracellular Ca^2+^ concentration ([Ca^2+^]_i_) and cell volume and how EwS cells cope with osmotic stress is unknown.

Generally, cellular Ca^2+^ and volume homeostasis are tightly regulated. Spatially and temporally coded miniscule changes in [Ca^2+^]_i_ and volume can induce and fine-tune various elements of cell physiology including cell migration, viability and proliferation. Cancer cells can differentially regulate ion channels involved in these procedures as means for developing (chemo-)therapy and apoptosis resistance and enhancing their invasiveness and metastatic potential [7]. Amongst the array of transporters and channels involved in these processes, Ca^2+^-dependent K^+^ channels stand out because of their versatility, playing a role in both Ca^2+^ homeostasis and volume regulation. These channels open upon an increase in the intracellular Ca^2+^ concentration and the resulting K^+^ efflux augments further Ca^2+^ influx by providing a favorable electrical gradient. Moreover, as K^+^ is osmotically active, cells utilize K^+^ efflux as means for counterbalancing cell swelling in a procedure termed regulatory volume decrease (RVD) [8]. 

## 2. Materials and Methods

### 2.1. Cells and Growth Conditions

Ewing Sarcoma, neuroblastoma and osteosarcoma cell lines were cultured in RPMI 1640 medium (Biochrom GmbH, Berlin, Germany) supplemented with 10% FCS and 1% glutamine (Gibco, Waltham, MA, USA) using 200µg/mL collagen G (Biochrom, Schaffhausen, Switzerland)-coated flasks at 37 °C, 5% CO_2_ for a maximum of 20 passages. For cell volume measurements, cells were incubated in HEPES-buffered RPMI 1640 medium. Table 1 contains a summarized list of cell lines used in this study. 

### 2.2. RNA Isolation, RT-PCR, RT-qPCR and Capillary Sequencing

RNA isolation from EwS cell lines and reverse transcription were performed using TRIzol^®^ (Thermo Fisher Scientific, Waltham, MA, USA) and SuperScript^®^IV Reverse Transcriptase (Invitrogen, Carlsbad, CA, USA), respectively, following the manufacturer’s protocol. RT-PCR and RT-qPCR primers are summarized in Appendix A. Exon-detecting primers for *KCNN1* are visualized in Appendix A. Primers were validated from previous publications when possible, or were designed de novo using Primer-BLAST [9]. 

RT-PCR was performed with the Red HS Master Mix (Biozym, Hessisch Oldendorf, Germany) using a Mastercycler^®^ (Eppendorf, Hamburg, Germany). Subsequently, gels of interest were cut out and DNA was extracted using the QIAquick^®^ Gel Extraction Kit (Qiagen, Hilden, Germany) following the manufacturer’s protocol. Subsequently, capillary sequencing was performed using Mix2Seq overnight sequencing kits (Eurofins Scientific, Luxembourg, Luxembourg) with sequencing primers highlighted in Appendix A.

RT-qPCR was performed with the PowerUp™SYBR^®^Green Mastermix (Thermo Fisher Scientific, Waltham, MA, USA) using a QuantStudio™ 3 cycler (Thermo Fisher Scientific, Waltham, MA, USA) and analyzed by the QuantStudio™ Design & Analysis Software (Thermo Fisher Scientific, Waltham, MA, USA). For RT-qPCR quantification, we normalized the gene expression of target genes to the mean expression of the housekeeper genes GAPDH and YWHAZ.

### 2.3. RNA In Situ Hybridization of Patient Tissue Microarrays

RNA in situ hybridization was performed on anonymized patient-derived TMAs derived from the cohorts Eicess92, E99 and EWING2008 [10,11], using custom-designed RNAScope™ probes (ACD Bio, Newark, CA, USA) and an RNAScope™ 2.5 HD Assay kit (ACD Bio, USA), according to the manufacturer´s protocol. Briefly, paraffin-embedded TMAs were deparaffinized and treated with hydrogen peroxide before antigen retrieval. Next, tissue slices were treated with protease, followed by hybridization and signal amplification. Finally, staining was visualized using a brightfield and fluorescence microscope (Zeiss Axiovert 100, Zeiss, Oberkochen, Germany). Each TMA was scored in a blinded manner by three individual researchers from 0 to 4 with the scoring algorithm detailed in Appendix A.

Afterwards, the mean scores were calculated and statistical analyses were performed with SAS. Correlation of dichotomized *KCNN1* with clinical parameters were computed with contingency coefficient (*r* = C). To analyze bivariate correlations, Spearman, Pearson or exact Fisher Test correlation coefficients were calculated. For survival analysis, overall survival was described by Kaplan-Meier curves, and survival functions between the groups were analyzed by a log-rank test. Fischer’s exact test was used to calculate *p*-values. *p*-values < 0.05 were considered as statistically significant. *p*-values were calculated from two-sided statistical tests, if not otherwise specified in the figure legends.

### 2.4. KCNN1 Knockdown

TC-71 cells, reaching a confluence of approximately 70%, were transfected with either *KCNN1* siRNA (ON-TARGETplus siRNA, Dharmacon, Lafayette, CO, USA) or scrambled siRNA (AllStars Negative Control siRNA, Qiagen, Hilden, Germany) using Lipofectamine RNAiMAX Reagent (Invitrogen, Carlsbad, CA, USA), following manufacturer´s protocol. Regulatory volume decrease measurements were performed on siRNA-treated cells 48 h after transfection. Transfection efficiency and specificity were controlled by RT-qPCR 24 h following transfection (Appendix A). Notably, siRNA transfection led to a 75% decrease in *KCNN1* expression (relative *KCNN1* expression compared to siScrambled transfection: 24.97 ± 2.61%, N = 3, n = 12). 

### 2.5. KCNN1 and EWSR1-FLI1 Overexpression

HEK-293 cells, endogenously not expressing *KCNN1* and EWSR1-FLI1, were subjected to transient transfection with pcDNA3-eGFP (#13031, Addgene, Watertown, MA, USA), pcDNA3-*KCNN1* (gifted by Prof. Dr. Neil V. Marrion, University of Bristol, Bristol, UK) or pCI-EWSR1-FLI1 and pCI-eGFP (both gifted by Jeff Toretsky, Georgetown University Washington, DC, USA) using Lipofectamine 3000 Reagent (Invitrogen, Carlsbad, CA, USA). Transfection efficiency was controlled by RT-qPCR and patch-clamp experiments were carried out 48 h following transfection. 

### 2.6. Analysis of RNA-Sequencing and ChIP-Sequencing Data

All analyses were run using suitable standard values for the below tools, unless stated otherwise. RNA-sequencing datasets GSE61950, GSE94277 and GSE160898 were downloaded from the NCBI SRA (Sequence Read Archive) using sratoolkit, quantified using Salmon [12], and imported into R using tximport [13]. Differential gene expression was analyzed with quasi-likelihood F-tests based on generalized linear models using edgeR [14]. Graphics were generated using ggplot2 [15]. ChIP-sequencing data from GSE61953 and from GSE101646 [16] were downloaded from the NCBI SRA, trimmed and mapped to hg38 using bbtools [17]. MACS2 callpeak and bdgcmp [18] were used to call peaks and generate fold-enrichment tracks over input, bedtools intersect [19] to filter backlisted regions [20]. Locations of GGAA microsatellites were predicted using Tandem Repeats Finder [21,22]. Graphics were generated using SparK [23].

### 2.7. Patch-Clamp Electrophysiology

For electrophysiolocal recordings, cultured cells were seeded on sterile round coverslips. Whole-cell recordings were performed at a conventional patch-clamp setup equipped with an EPC-10 amplifier and PatchMaster software (HEKA Elektronik, Lamprecht, Germany). Recording pipettes were fabricated from borosilicate glass (G150TF-10, Clark Electromedical Instruments, Pangbourne, UK). Typical electrode and series resistances were 3–4 MΩ and 6–10 MΩ, respectively. Series resistance compensation of ≥30% was routinely applied. Once the whole-cell configuration was achieved, the pipette solution and intracellular milieu equilibrated within 1 min. Cells were recorded at RT, held at a potential of −40 mV and membrane outward currents were evoked using a ramp protocol (5 s duration) ranging from −80 to +60 mV. The current amplitudes at −50 mV of the depolarizing ramp were analyzed. A liquid junction of 21 mV was taken into consideration. Test substances were applied in close proximity to the recorded cell using a multibarrel application pipette with a diameter of about 100 µm. The following recording solutions were used: (1) Extracellular solution (in mM): 160 Na-gluconate, 5 KCl, 10 HEPES, 1 MgCl_2_, 1 CaCl_2_; pH and osmolality were set to 7.3 and 310 mOsmol/kg, respectively. (2) Intracellular solution containing 5 µM free Ca^2+^ (in mM): 145 K-gluconate, 10 HEPES, 1.3 EGTA, 1 MgCl_2_, 1.217 CaCl_2_, 1 Mg-ATP; pH and osmolality were set to 7.2 and 295 mOsmol/kg, respectively.

### 2.8. Intracellular Ca^2+^ Measurement

Prior to the measurements, approximately 40,000 Ewing sarcoma cells were left to adhere on glass-bottom dishes (MatTek Life Sciences, Ashland, MA, USA) coated with 1 mg/mL fibronectin (Roche, Rotkreuz, Switzerland) at 37 °C and 5% CO_2_ overnight. Next, cells were stained with 6 µM of the ratiometric Ca^2+^ indicator dye Fura-2-AM (Cayman Chemical, Ann Arbor, MI, USA) at 37 °C for 30 min and subsequently washed with HEPES-buffered Ringer solution (NaCl: 122.5 mM, KCl: 5.4 mM, CaCl_2_: 1.2 mM, MgCl_2_: 0.8 mM, glucose: 10 mM, HEPES: 10 mM). Subsequently, an ionic imaging setup (Visitron, Puchheim, Germany) consisting of an inverted fluorescence microscope (Zeiss Axiovert 100, Zeiss, Oberkochen, Germany), an integrated high-speed shutter system, a heated perfusion system and a polychromator was used to monitor intracellular Ca^2+^ dynamics at 37 °C. During the measurement, cells were superfused initially with HEPES-buffered Ringer solution, followed by Ca^2+^-free Ringer solution (NaCl: 122.5 mM, KCl: 5.4 mM, MgCl_2_: 0.8 mM, glucose: 10 mM, EGTA: 1 mM) to eliminate extracellular Ca^2+^. Then, Ca^2+^-free Ringer solution was supplemented with 1 µM thapsigargin (Böhringer, Ingelheim, Germany) to allow emptying of intracellular Ca^2+^ stores. Next, cells were superfused with Ca^2+^ containing HEPES-buffered Ringer solution with 1 µM thapsigargin to induce SOCE in the presence or absence of the K_Ca_2 channel modulator 100 nM apamin. Lastly, [Ca^2+^]_i_ was calibrated using Ca^2+^-free Ringer solution supplemented with the Ca^2+^ ionophore 1 µM ionomycin (AK Scientific, Union City, CA, USA), then 5 mM Ca^2+^-containing Ringer solution + 1 µM ionomycin. Images were acquired every 20 s using dual excitation with wavelengths of 340 nm (F_340_) and 380 nm (F_380_), corresponding to the excitation maxima of the Ca^2+^-bound and Ca^2+^-free form of the fluorophore, respectively, and emission was detected at 520 nm. F_340_/F_380_ ratio was derived from these values in each cell over time and [Ca^2+^]_i_ was calculated as described by Grynkiewicz et al. [24]. 

### 2.9. Migration Assays

To assess the two-dimensional random migration of Ewing sarcoma cells, 10^5^ cells in cell culture medium were seeded onto 12.5 cm^2^ flasks coated with an extracellular matrix composed of 89% collagen I (Corning Inc., Corning, NY, USA), 4.5% laminin (Sigma-Aldrich, St. Louis, MI, USA), 4.5% fibronectin (Roche, Rotkreuz, Switzerland), 1.3% collagen III (Beckton Dickinson, Franklin Lakes, NJ, USA) and 0.6% collagen IV (Corning Inc., Corning, NY, USA). After overnight incubation at 37 °C and 5% CO_2_, flasks were supplemented with modulators depending on experimental condition, namely: 10 ng/mL TGF-β (Peprotech, Rocky Hill, KT, USA), 100 nM apamin (Alomone, Jerusalem, Israel) or 1 µM NS8593. Next, the pH of the medium was equilibrated at 37 °C and 5% CO_2_ for an additional 30 min, after which the flasks were shut tight and were placed into 37 °C temperature-controlled chambers under phase-contrast microscopes (Zeiss Axiovert 40C, Zeiss, Oberkochen, Germany). Time-lapse series of two-dimensional cell migration were recorded over 6 h at a time interval of 15 min using a CMOS camera (MikroCam SP 3.1, Bresser, Rhede, Germany) with the compatible acquisition software (MicroCamLab, Bresser, Rhede, Germany). Cells were segmented using the Amira software (Thermo Fischer Scientific, Waltham, MA, USA), and ultimately, velocity and start-to-end translocation of individual cell trajectories were derived as described previously [25,26]. 

Spheroid invasion assays with Ewing sarcoma and osteosarcoma spheroids were performed as previously described [25]. First, cells after trypsinization were counted and resuspended in a medium with 0.24% methyl cellulose (Sigma-Aldrich, St. Louis, MO, USA). Next, droplets of 50 µL, containing 3000 cells each, were allowed to form spheroids as hanging drops in a 10 cm Petri dish at 37 °C and 5% CO_2_ for 48 h. Afterwards, spheroids were embedded into 100 µL droplets of collagen I-based extracellular matrix with the composition detailed above in 12.5 cm^2^ flasks. After matrix polymerization at 37 °C and 5% CO_2_ for 2 h, cell culture medium supplemented with 1µg/mL IGF-1 (Peprotech, Rocky Hill, KT, USA) to stimulate matrix invasion of sarcoma cells. Furthermore, medium was supplemented with either 0.1% DMSO for control or with 100 nM Apamin (Alomone, Jerusalem, Israel) and 10 µM Senicapoc (gifted by Prof. Dr. Heike Wulff, University of California Davis) for K_Ca_ channel inhibition. After pH equilibration of the medium described above, spheroid invasion was recorded at 37 °C using phase-contrast microscopy over 48 h at a time interval of 10 min. Analysis was carried out using ImageJ. The spheroid’s size over time was measured every hour for the whole observation period. In the first 8 h, the SOAS exercised a visible pull to the matrix, which was additionally measured every ten minutes.

### 2.10. Cell Viability Assays

Assessment of Ewing Sarcoma cell viability treated with K_Ca_ channel inhibitors in the presence of chemotherapeutic drugs was performed using the MTT assay. Briefly, 27,500 cells were seeded into wells of a 96-well plate and after overnight incubation at 37 °C, 5% CO_2_, cells were treated with chemotherapeutic drugs routinely used in the treatment of Ewing sarcoma, namely, doxorubicin and/or vincristine (Sigma-Aldrich, St. Louis, MI, USA). Moreover, the effect of chemotherapy was investigated in the presence or absence of the K_Ca_2.1 inhibitor apamin. Treatment duration was 72 h, after which cells were treated with 0.5 mg/mL MTT reagent (3-(4,5-dimethylthiazol-2-yl)-2,5-diphenyl tetrazolium bromide, Sigma-Aldrich, St. Louis, MI, USA) at 37 °C, 5% CO_2_ for 1 h. Next, plates were centrifuged at 300 g for 5 min, and the solution was exchanged to 100 µL DMSO in each well. MTT absorbance was detected using a spectrophotometer (THERMOmax, Beckman Coulter, Brea, CA, USA) at wavelengths of 546 nm, as well as 650 nm as a reference wavelength. Data were normalized to the absorbance of the control treated with 0.1% DMSO. Each set of experiments was performed with technical triplicates.

Ewing sarcoma and osteosarcoma cell survival after hypoosmotic shock treatment was performed using the sulforhodamine B (SRB) assay. To examine the osmotic resistance of cells, a series dilution was performed by mixing cell culture medium with hypotonic solution (26.8 mOsm) containing 1.2 mM CaCl_2_, 0.8 mM MgCl_2,_ 5.4 mM KCl, and 10 mM HEPES. For the nominally zero mOsm solution, dH_2_O was used. Prior to the experiment, 20,000 cells were seeded into wells of a 96-well plate coated with 25 µg/mL fibronectin at 37 °C, 5% CO_2_ for 24 h. Next, cell culture medium was exchanged to solutions with different osmolarities, namely 279 mOsm, 153 mOsm, 111 mOsm, 77 mOsm, 55 mOsm, 27 mOsm and 0 mOsm, for 5 min. After 5 min, wells were washed once with the same solution to eliminate dead cells that lost adherence. Afterwards, SRB assay was performed as described by Orellana and Kasinki [27]. Briefly, cells in each well were fixed using trichloracetate at 4 °C for 60 min. Then, the wells were washed four times with dH_2_O, after which fixed cells were stained with 0.04% SRB (Sigma-Aldrich, St. Louis, MO, USA) solved in 1% acetic acid at RT for 1 h. Next, wells were washed four times with 1% acetic acid, then were left to air dry for an additional 30 min. Lastly, SRB-protein complexes in each well were dissolved using Tris-Base-solution (10 mM, pH 10.5). SRB absorbance was detected with the same settings as MTT absorbance (see above). Data were normalized to the absorbance of control treated with the isosmotic (279 mOsm) solution. Each set of experiments was performed with technical triplicates. 

### 2.11. Cell Volume Measurements

For cell volume measurements by fluorescence exclusion microscopy [28], cells were seeded into Luer µ-Slides (Ibidi, Gräfelfing, Germany) coated with Fibronectin (25 µg/mL; Roche, Rotkreuz, Switzerland) and left to adhere in HEPES-buffered RPMI 1640 medium at 37 °C overnight. Next, the Luer µ-Slides were superfused with solutions containing 400 µg/mL 70 kDa-FITC-dextran using a syringe and visualized using a confocal microscope (Lecia DMI 6000, Leica, Mannheim, Germany). Because of the high molecular mass, FITC-dextran does not penetrate the cell membrane, so we obtained negative images of the cells. To obtain a good approximation of cellular volume, z-stacks of 0.3 µm resolution were taken, and measured cell volume was calculated by performing a volume rendering with Amira 2019.2 (Thermo Fisher Scientific, Waltham, MA, USA). The measured volumes were multiplied by 0.83 because of an aberration induced by the different refractive indices of the immersion oil and the Ringer solution [29]. During the measurement, cells were firstly incubated in isosmotic HEPES-buffered Ringer solution (271.8 mOsm, composition see above), and after 5 min, medium was exchanged with an isosmotic solution where NaCl was partly replaced by mannitol. By means of the mannitol equilibration, it was ensured that cell volume changes were not induced by single electrolyte concentrations, but only by the osmolarity. Next, a hypoosmotic solution without mannitol (171.8 mOsm or 221.8 mOsm) was used to induce cell swelling and subsequently RVD in the presence or absence of the K_Ca_2.1 inhibitor 100 nM apamin. Z-stacks were acquired every 2 min for 20 min. Analysis was carried out using the Amira software. Cell volumes of every cell were normalized to the initial cell volumes in the mannitol-equilibrated isosmotic solution.

## 3. Results

### 3.1. Multiple Ca^2+^-Modulated Ion Channels Are Functional in Ewing Sarcoma

Osteolytic growth of EwS is coupled with marked local Ca^2+^ release from the degraded bone. We hypothesized that tumor cells adapt to the increased [Ca^2+^]_e_ by expressing Ca^2+^-regulated ion channels (Figure 1A). Multiple Ca^2+^-modulated ion channels are involved directly or indirectly in store-operated entry (SOCE). Intracellular Ca^2+^ homeostasis as well as the relevant Ca^2+^ regulatory ion channels in EwS have not yet been studied. We conducted intracellular Ca^2+^ measurements in cells from two established EWS cell lines (Figure 1B–D), where we found that both SK-ES-1 and A673 cells have a resting [Ca^2+^]_i_ ranging from 100 to 200 nM (Figure 1C; SK-ES-1: 174 ± 9 nM Ca^2+^,N = 4, n = 108; A673: 132 ± 9 nM Ca^2+^, N = 4, n = 111). Moreover, EwS cells contain functional Ca^2+^ stores and utilize SOCE triggered by thapsigargin (TG) stimulation (Figure 1D, peak [Ca^2+^]_i_ concentrations of SK-ES-1: 576 ± 26 nM Ca^2+^, N = 4, n = 108, *p* < 0.0001; A673: 266 ± 18 nM Ca^2+^, N = 4, n = 111, *p* < 0.0001). 

To gain a better understanding of the potential molecular players of Ca^2+^ signaling in EwS cells, we inspected the expression of [Ca^2+^]_i_ regulating ion channel genes in a published EwS RNAseq surfaceome database [30]. We selected 22 highly expressed genes from the TRP, Stim/Orai, K_Ca_, Ca_v_ and Piezo families of ion channels and confirmed their expression using RT-qPCR in three EwS cell lines, namely, SK-ES-1, TC-71 and A673 (Figure 1E, expression relative to *GAPDH* and *YWHAZ*, N = 3, n = 6). Seven of the twenty-two genes are expressed at a high level in all three EwS cell lines, namely *TRPM7*, *TRPC1*, *STIM2*, *ORAI1*, *ORAI2*, *KCNN1* and *PIEZO1*. Taken together, we found that multiple ion channels possibly involved in intracellular Ca^2+^ homeostasis and SOCE are present in EwS cells. Based on its particularly high abundance and its unique biophysical and pharmacological properties we decided to focus on *KCNN1* and its protein product, the K_Ca_2.1 channel, in this study. 

### 3.2. EWSR1-FLI1 Drives the Expression of an Alternative KCNN1 Transcript Variant in Ewing Sarcoma

To investigate whether *KCNN1* has a specificity for EwS over the alternative primary bone sarcoma in this age group, osteosarcoma, we performed RT-qPCR examining a larger panel of EwS cell lines (SK-ES-1, A673, TC-71, 5838, TTC-466 and CADO) and compared *KCNN1* expression to the osteosarcoma cell lines HOS, SAOS and ZK-58 (Figure 2A). We found that osteosarcoma cell lines have negligible *KCNN1* expression compared to EwS cell lines (expression relative to *GAPDH* and *YWHAZ*: EwS: 0.02 ± 0.003, N = 3, n = 48; OS: 0.0001 ± 0.00002, N = 3, n = 22). Furthermore, we found out that *KCNN1* is expressed in tumor cells but not in the surrounding healthy bone tissue using RNA in situ hybridization (RNAish) (Figure 2B, N = three whole EwS tissue slides). Overall, these findings indicate that *KCNN1* has a considerable tissue specificity in EwS compared to osteosarcoma and healthy bone tissue.

Next, we analyzed whether EWSR1, when fused to ETS family transcription factors such as FLI1, drives the expression of *KCNN1*. As a proof of principle, we focused on the fusion oncogene EWSR1-FLI1, and evaluated whether it directly or indirectly affects *KCNN1* expression. RNAseq datasets (GSE27524 and GSE103837, respectively) from two independent reports by Bilke et al. [31] and by Zöllner et al. [32] show that *KCNN1* is downregulated upon *EWSR1-FLI1* knockdown. We reanalyzed the ChiPseq datasets GSE101646 and GSE61953 [16,33] and found that EWS-FLI1 establishes an enhancer at the *KCNN1* locus. H3K4me1 (mono-methylation at the fourth lysine residue of the histone H3 protein) marks two poised enhancers at the *KCNN1* locus that are inactive in HEK293 cells. FLI1 directly binds to an alternative enhancer in the third intron of *KCNN1* (“Enhancer B” in Figure 2C), which consists of GGAA microsatellites. This leads to a strong activation of this enhancer in EwS cells, as evidenced by increased H3K27ac (acetylation of the lysine residue at N-terminal position 27 of histone H3). Moreover, around the GGAA microsatellite, the enhancer (“Enhancer B”) signal becomes much broader. Activation of this enhancer leads to transcriptional activation of the *KCNN1* gene (as visualized by H3K4me3 (tri-methylation at the fourth lysine residue of the histone H3 protein)) in the EwS cell lines A673 and SKMNC but not in HEK293 lacking FLI1 activity. Overall, the differential methylation and acetylation patterns indicate that the usage of an alternative enhancer facilitates the transcription of *KCNN1*. As a further link between EWSR1-FLI1 and *KCNN1*, we found a more than two-fold higher expression of *KCNN1* in *EWSR1-FLI1*-transfected HEK293 cells compared to control transfection (Figure 2D, expression relative ctrl transfection, ctrl vs. *EWSR1-FLI1* transfection: 1.00 ± 0.07, N = 3, n = 12; vs. 2.17 ± 0.24, N = 3, n = 9, *p* < 0.0001). Published datasets from GSE160898 [34] confirm that overexpression of EWSR1-FLI1 induces *KCNN1* expression in HEK293 cells (Figure 2E). In contrast, when EWSR1-FLI1 is either inhibited pharmacologically or knocked down in A673 and SKMNC, the expression of *KCNN1* drops significantly (Figure 2E, data from GSE94277 and GSE61950, respectively [33,35]).

After finding that EwS cells utilize a non-canonical enhancer, we investigated whether this results in the expression of non-canonical *KCNN1* transcript variants. Besides the canonical transcript (NCBI RefSeq: NM_002248), multiple other transcript variants can be found in the NCBI database. *KCNN1* has mutually exclusive exons, with the canonical transcript utilizing exons 1 and 2, but not exon 3, whereas multiple alternative transcripts (such as NM_001386974) utilize exon 3 but not exons 1 and 2 (Figure 3A). The downstream 3’ exons 4–12 of each transcript variant are highly similar. To dissect the quantitative expression of different exons, we utilized RT-qPCR with exon-specific primers (Figure 3B). Indeed, exons 1 and 2, and exon 3 have similar expression levels in EwS cell lines A673, SK-ES-1 and TC-71 (expression relative to *GAPDH* and *YWHAZ* Exon 1: 0.0001 ± 0.00002, N = 3, n = 14; Exon 2–4: 0.00005 ± 0.00002, N = 3, n = 14; Exon 3–4: 0.00214 ± 0.001, N = 3, n = 14). Moreover, 3’ exons (Exon 5–6: 0.12 ± 0.004, N = 3, n = 14) show a higher expression than exons 1 (*p* = 0.0025), 2 (*p* = 0.0025) and 3 (*p* = 0.01). 

To verify that the above results are not due to the lack of primer specificity, we amplified longer sequences of canonical and alternative transcripts (Figure 3C). We found amplicons corresponding to the expected sizes of the canonical as well as the alternative transcript variant 2 (NM_001386974) in all EwS cell lines (SK-ES-1, A673, TC-71, CADO, 5838 and TTC-466) but not in the osteosarcoma lines HOS, SAOS or ZK-58 (N = 3, n = 3). 

Surprisingly, we also found a short transcript when amplifying canonical exons 1–8. To assess the sequence of the short transcript, and to confirm the sequence of the other transcripts, we resorted to capillary sequencing (Figure 3D, Suppplementary Data). This confirmed the presence of the canonical transcript (NM_002248) and the alternative transcript variant 2 (NM_001386974). We termed the previously not characterized transcript variant 2 of *KCNN1* driven by EWSR1-FLI1 *KCNN1_EWS_*. As exon 3 is upstream of the open reading frame (ORF), the protein sequence resulting from *KCNN1_EWS_* translation does not change compared to the canonical K_Ca_2.1. Sequencing also revealed that the short transcript is due to splicing out exons 5–6. This results in the lack of the S2 and S3 transmembrane domains of the resulting K_Ca_2.1 channel protein, likely rendering the product non-functional.

### 3.3. KCNN1 Expression Does Not Lead to K_Ca_2.1 Function in Ewing Sarcoma 

Based on the above results (Figure 2B), we tested whether KCNN1 could be a feasible diagnostic or prognostic marker in EwS. As available antibodies against K_Ca_2.1 are not specific (Appendix A), we used a human *KCNN1*-specific 20ZZ probe (targeting nucleotides 1153–2659 recognizing all three described transcripts) for RNAish in tissue microarrays of 217 EwS patients from cohorts Eicess92, E99 and EWING2008 [36,37,38] (Figure 4A). 

There was a substantial heterogeneity in the RNAish signal of EwS patients and we clustered 78% of the patients into the *KCNN1*^high^ group, whereas RNAish signal was scarce in 22% of EwS patients (*KCNN1*^low^ group) (Figure 4A,B). In a double-blinded experiment, we correlated RNAish signal intensity with various aspects of clinical outcome such as overall survival (Figure 4C). We found no correlation with overall survival (hazard ratio and 95%CI: 0.85 (0.45–1.61), *p* = 0.62) and with other clinical parameters such as tumor volume, age at diagnosis, presence of primary metastases, therapy response, and time to first relapse (summarized in Appendix A). In summary, quantitative *KCNN1* expression assessed using RNAish is not a suitable prognostic marker. 

### 3.4. K_Ca_2.1 Is Non-Functional in EWS Cells 

Having ruled out the possibility of using *KCNN1* as a prognostic marker, we tested whether the druggability of the protein product of *KCNN1* could be potentially exploited for EwS therapy (Figure 5). K_Ca_2.1 can be selectively inhibited by the bee toxin apamin and has a very restricted tissue distribution [39], making it an attractive therapeutical target. We therefore addressed the impact of K_Ca_2.1 function on numerous other aspects of EwS cell behavior (Figure 5A). Our cell migration studies on a collagen-coated two-dimensional substrate (Figure 5B) indicate that cell migration of EwS cells treated with 10 ng/mL TGF-β is not influenced by the presence of 100 nM apamin ( migration velocities of control vs. apamin; A673: 0.29 ± 0.02 µm/min vs. 0.23 ± 0.02 µm/min, N = 4, n = 40, *p* = 0.25; SK-ES-1: 0.34 ± 0.02 µm/min vs. 0.31 ± 0.02 µm/min, N = 5, n = 48, *p* = 0.85; TC-71: 0.28 ± 0.02 µm/min vs. 0.24 ± 0.02 µm/min, N = 4, n = 38, *p* = 0.72). We considered the possibility that K_Ca_2.1 might be expressed intracellularly, e.g., in mitochondria, similar to other K_Ca_ channels [40]. Therefore, we inhibited K_Ca_2.1 also with the membrane-permeable inhibitor NS8593 (Appendix A) [41]. We found that 1 µM NS8593 does not affect EwS cell migration (migration velocities of control vs. NS8593; A673: 0.31 ± 0.05 µm/min vs. 0.29 ± 0.04 µm/min, N = 3, n = 30, *p* = 0.99; SK-ES-1: 0.14 ± 0.02 µm/min vs. 0.16 ± 0.02 µm/min, N = 4, n = 40, *p* = 0.99). 

To test whether K_Ca_2.1 is involved in EwS cell migration in a more complex system, we created spheroids from SK-ES-1 cells, embedded them into a complex collagen-based extracellular matrix similar to the bone [5] and assessed the motility of EwS cells within these structures (Figure 5C). Inhibition of both K_Ca_2.1 and K_Ca_3.1 by using apamin and senicapoc in combination failed to attenuate EwS cell motility out of the spheroid (relative SK-ES-1 spheroid area @48 h control vs. apamin+senicapoc: 2.14 ± 0.13 vs. 2.50 ± 0.26 µm/min, N = 4, n = 8, *p* = 0.25). Generally, we found that SK-ES-1 cells are not particularly motile in a three-dimensional environment (Appendix A). For comparison, we quantified their motility along with cells from an osteosarcoma cell line (SAOS, Figure 5D,E, Appendix A). SAOS osteosarcoma cells are vividly emigrating out of the spheroid into their surroundings (relative spheroid area @48 h SK-ES-1 vs. SAOS: 2.14 ± 0.13 vs. 3.30 ± 0.54 µm/min, N = 4, n_SK-ES-1_ = 8, n_SAOS_ = 7, *p* = 0.03). In summary, EwS cell migration in a two-dimensional as well as in a three-dimensional system both remain unaltered upon K_Ca_2.1 modulation.

K_Ca_ channel function has previously also been implicated in the sensitization of cancer cells to chemotherapy [42,43]. Thus, we investigated whether EwS cell responsiveness to chemotherapy is affected by K_Ca_2.1 inhibition (Figure 5F,G). We applied two routinely used chemotherapeutic agents with different mechanisms of action, namely doxorubicin and vincristine. Our results show that 100 nM apamin does not improve the chemosensitivity of EwS cells to chemotherapeutic drugs; in fact, we observed increased chemoresistance upon apamin application in both A673 and SK-ES-1 cells (Figure 5G, normalized survival 100 nM doxorubicin vs. 100 nM doxorubicin + 100 nM apamin A673: 0.14 ± 0.01 vs. 0.23 ± 0.02, N = 3, n = 6, *p* = 0.009; SK-ES-1: 0.22 ± 0.02 vs. 0.37 ± 0.01, N = 3, n = 6, *p* < 0.0001; 10 nM vincristin vs. 10 nM vincristin + 100 nM apamin A673: 0.05 ± 0.01 vs. 0.10 ± 0.01, N = 3, n = 6, *p* = 0.0002 SK-ES-1: 0.07 ± 0.01 vs. 0.08 ± 0.01, N = 3, n = 6, *p* = 0.51).

To further confirm the lack of K_Ca_2.1 function in EwS cells, we tested whether 100 nM apamin has any effects on Ca^2+^ homeostasis (Figure 5H). As expected, we found that apamin has no effect on SOCE in both A673 and SK-ES-1 cells (Figure 5I, peak [Ca^2+^]_i_ of control vs. 100 nM apamin A673: 199 ± 14 nM vs. 199 ± 17 nM, N = 3, n = 62, *p* = 0.99; SK-ES-1: 450 ± 22 nM vs. 411 ± 17 nM, N = 4, n = 137, *p* = 0.15). 

Overall, the results from a variety of functional in vitro assays (Figure 5) consistently argue against any functional role of K_Ca_2.1 in EwS cell lines despite its overexpression. This is consistent with the fact that *KCNN1* expression does not correlate with the clinical parameters of EwS (Figure 4).

### 3.5. KCNN1 Transcripts in EwS Are Non-Conductive

Our results outline a discrepancy between very robust *KCNN1* expression derived from molecular biology (Figure 1, Figure 2 and Figure 3) and a lack of ion channel function, as derived from our clinical prognostic and functional cellular analyses (Figure 4 and Figure 5). To resolve the discrepancy, we hypothesized that *KCNN1* transcripts in EwS conduct no K^+^ and thereby carry no conductive role in EwS cell physiology. To confirm this idea, we utilized the gold-standard assay for ion channels, whole-cell patch-clamp (Figure 5A–C). In SK-ES-1 and TC-71 cells, membrane currents were generally small and revealed a rather linear current-voltage relationship reversing at around −33 mV, thereby indicating a deficiency in functional K^+^ channels. As expected, we found that 100 nM apamin has no effect on ionic currents in EwS cells (Figure 6A,C; currents at −50 mV control vs. 100 nM apamin; SK-ES-1: −35 ± 10 pA, versus −50 ± 14 pA, N = 5, n = 12, *p* = 0.38; TC-71: −44 ± 18 pA versus −38 ± 14 pA, N = 5, n = 12, *p* = 0.81). Strikingly, EwS cells lack outward current even at −50 mV, indicating that they lack any K_Ca_ function. In addition, the current slope is largely linear, indicating a deficiency in K_V_ channels and, generally, K^+^ conductance. Notably, we found that the reversal potential of EwS cell lines is ≈ −33 mV at the given ionic composition of the medium. Thus, the background ionic conductance of the EwS cell membrane is very likely due to Cl^−^, that has an equilibrium potential of −30 mV. 

Next, we tested whether the absence of K_Ca_2.1 function can possibly be explained by a dominant negative effect of the short *KCNN1* transcript or *KCNN1*_EWS_. We hypothesized that these variants, similarly to K_Ca_3.1, could form heteromers with K_Ca_2.1 [44], rendering the channel non-functional. To this end, we transfected HEK293 cells with the canonical *KCNN1* and/or *EWSR1-FLI1* (Figure 6B). HEK293 cells, transiently transfected with *KCNN1*, expressed apamin-sensitive K_Ca_2.1 currents, verifying our earlier methodology (currents at −50 mV HEK-*KCNN1* control vs. 100 nM apamin: 435 ± 134 pA versus 39 ± 30 pA, N = 3, n = 7, *p* = 0.01). In contrast, HEK293 cells, transfected solely with EWSR1-FLI1, expressed no apamin-sensitive K_Ca_2.1 currents (currents at −50 mV HEK-*EWSR1-FLI1* control vs. 100 nM apamin: −46 ± 9 pA, −45 ± 9 pA, N = 3, n = 7, *p* = 0.95), even though they expressed *KCNN1* (Figure 2D). When co-transfecting HEK293 cells with *EWSR1-FLI1* and *KCNN1*, we observed similar currents as with single *KCNN1* transfection (currents at −50 mV HEK-*KCNN1* vs. HEK-*EWSR1-FLI1*-*KCNN1*: 435 ± 134 pA versus 360 ± 56 pA, N = 3, n = 7, *p* = 0.63). This evidence argues against a dominant negative effect of alternative *KCNN1* transcripts in EWS. During these experiments, we recorded the resting membrane potential (RMP) of the measured cells (Figure 6D). EwS cell lines have a markedly depolarized membrane potential near the equilibrium potential of Cl^−^ (SK-ES-1: −24 ± 3 mV, TC−71: −23 ± 3 mV), which is a logical consequence of their absent K^+^ conductance. In contrast, the resting membrane potential of HEK293 cells transfected with the canonical K_Ca_2.1 is largely hyperpolarized (HEK-*KCNN1*: −78 ± 3 mV, HEK-*EWSR1-FLI1-KCNN1*: −67 ± 9 mV). In summary, EWSR1-FLI1-induced pathognomonic *KCNN1* variants are non-functional transcripts that likely do not exert a dominant negative effect on canonical K_Ca_2.1 channel function.

### 3.6. Lack of K_Ca_2.1 Channel Function Sensitizes EwS Cells to Hypoosmotic Stress 

Throughout our study, the absence of any detectable K_Ca_2.1 channel activity, and more broadly, K^+^ channel function in EwS cells was perplexing. Despite the negative results, we saw this as an opportunity to selectively target EwS cells. Generally, K^+^ conductance has a vital role in cell volume homeostasis. Thus, we hypothesized that the absence of K^+^ channel function hampers the defensive mechanisms of EwS cells to hypoosmotic stress. As a proof of concept, we applied hypoosmotic shock (170 mOsm) on EwS cells and observed a regulatory volume decrease (RVD) (Figure 7A,B) [8]. Commonly, eukaryotic cells can completely regenerate their original volume within 10–20 min after mild hypotonic shock. However, we found that TC-71 cells barely regulate their volume, as represented by a low overall RVD even after 20 min (Figure 7C). Moreover, we tested whether the inhibition or genetic knockdown of the K_Ca_2.1 affects the volume regulation of TC-71 cells. We found no significant differences in the cell volume between *KCNN1*-silenced cells and cells treated with scrambled control siRNA (Figure 7C), or upon 100 mM apamin treatment (Appendix A) at any time (relative cell volume 20 min after hypotonic shock, siScrambled versus si*KCNN1*: 1.34 ± 0.02 versus 1.34 ± 0.02, N = 4, n_siScrambled_ = 38, n_siKCNN1_ = 32, *p* = 0.86; control versus 100 nM apamin: 1.32 ± 0.02 versus 1.35 ± 0.03, N = 4, n_control_ = 28, n_apamin_ = 27, *p* = 0.26). 

To evaluate whether the poor RVD of the TC-71 is a general feature of EwS cells, the RVD-assays were repeated with a milder hypotonic shock (220 mOsm) and with an additional EwS cell line (SK-ES-1) as well as with an osteosarcoma cell line (SAOS) (Figure 7B). Compared to the osteosarcoma cell line SAOS, the EwS cell lines TC-71 and SK-ES-1 exhibit a substantially lower RVD (Figure 7D, relative cell volume 20 min after hypotonic shock, SAOS versus TC-71 versus SK-ES-1: 1.00 ± 0.02 versus 1.20 ± 0.0 versus 1.08 ± 0.02, respectively, N = 4, n_SAOS_ = 33, n_TC-71_ = 26, n_SK-ES-1_ = 42, *p*_SAOS-TC__—71_ < 0.0001, *p*_SAOS-SK-ES-1_ = 0.0001). Notably, in a 220 mOsm solution, the TC-71 cells do not exhibit any RVD at all, which is highly uncommon in eukaryotic cells (relative TC-71 cell volume, 2 min versus 20 min after hypotonic shock: 1.20 ± 0.01 versus 1.20 ± 0.01, N = 4, n = 26, *p* = 0.8). 

To verify whether this poor volume regulation of the EwS cells results in a vulnerability for hypotonic stress, EwS and osteosarcoma cells were exposed to different hypotonic solutions for 15 min, and cell survival was subsequently measured with a sulforhodamine B (SRB)-assay (Figure 7E). After incubation with hypotonic solutions of 77 mOsm or less, TC-71 EwS cell survival was considerably lower than that of SAOS osteosarcoma cells (relative SRB absorbance at 77 mOsm, TC-71 versus SAOS, 0.57 ± 0.07 versus 0.96 ± 0.03, N = 3, n = 9, *p* < 0.0001). The IC_50_ value of osmolarity-dependent survival of TC-71 was 65.7 mOsm. In sharp contrast, SAOS cells were still viable even after 15 min in distilled water (relative SRB absorbance after incubation in distilled water, TC-71 (Appendix A) versus SAOS (Appendix A) (0.01 ± 0.01 versus 0.59 ± 0.03, N = 3, n = 9, *p* < 0.0001). 

To test whether this effect is common in EwS cells compared to osteosarcoma cells, we repeated the assay for three additional EwS cell lines (SK-ES-1, A673 and TTC-466) and two additional osteosarcoma cell lines (HOS and ZK-58) at 55 mOsm, where the survival difference between TC-71 and SAOS cells was the highest (Figure 7F). Indeed, EwS cell lines were extremely sensitive to hypotonic shock, represented by a significantly lower SRB absorbance compared to the isotonic solution (SRB absorbance isotonic versus hypotonic, N = 3, n = 9, TC-71: 1.00 ± 0.04 versus 0.40 ± 0.04, *p* < 0.0001; SK-ES-1: 1.00 ± 0.01 versus 0.57 ± 0.02, *p* < 0.0001; A673: 1.00 ± 0.01 versus 0.85 ± 0.02, *p* < 0.0001; TTC-466: 1.00 ± 0.01 versus 0.51 ± 0.02, *p* < 0.0001). In contrast, two of three osteosarcoma cell lines were osmoresistant (SRB absorbance isotonic versus hypotonic, N = 3, n = 9, SAOS: 1.00 ± 0.02 versus 1.05 ± 0.03, *p* = 0.67; HOS: 1.00 ± 0.01 versus 0.95 ± 0.02, *p* = 0.52; ZK-58: 1.00 ± 0.01 versus 0.89 ± 0.01, *p* = 0.003).

## 4. Discussion

Our study aimed to better understand the complex Ca^2+^ homeostasis of an osteolytic tumor, EwS, and specifically to outline the “Ca^2+^ toolkit” necessary for EwS cells to cope with increased environmental [Ca^2+^]. Our work sheds first light on the expression of ion channel genes that may coordinate Ca^2+^ homeostasis in EwS. Among them, we found the expression of genes that encode channels permeable to Ca^2+^, such as SOCE-mediating channels, Piezo 1 and 2, as well as voltage-gated Ca^2+^ channels (e.g., Ca_V_3.1) and TRP channels (including TRPC1, TRPM7, and TRPV4). We also found prominent gene expression in the case of K^+^ channels that are activated by a rise in the intracellular Ca^2+^ concentration, specifically in *KCNN1* encoding K_Ca_2.1. To understand why *KCNN1* expression is prominent, we assessed the epigenetic landscape of the locus. Previously, it has been described that the promoters of multiple voltage-gated K^+^ channels’ genes are differentially methylated in EwS when compared with nonmalignant adult tissues [45]. Additionally, differential promoter methylation also causes epigenetic dysregulation in the *KCNN4* locus in lung cancer [46]. Here, we demonstrated a differential histone methylation and acetylation pattern (Figure 2) in the *KCNN1* locus with the consequence of activating an alternative enhancer repressed in healthy tissue and ultimately driving the expression of the alternative channel transcript that we termed *KCNN1_EWS_*. Whether other Ca^2+^ (regulated) ion channel genes outlined in our work could also have cryptic enhancers that become activated by the oncoprotein EWSR1-FLI1 remains to be studied. 

Throughout our study, we were confronted with the discrepancy that high *KCNN1* expression at the mRNA level in all the EwS lines studied as well as tumor cells in EwS patient material is by no means reflected in the functional assays and in the clinical data. The seemingly simple, and at the same time intriguing, explanation is that there is no K_Ca_2.1 channel-mediated current in Ewing sarcoma cells and, consequently, none of the functional assays revealed an impact of the channel on EwS cell behavior. We addressed this conundrum in multiple ways. In most assays, we used the bee venom apamin to inhibit K_Ca_2.1. Because apamin is a polypeptide acting from the extracellular side [47], we considered that the reduced effectiveness of apamin can be due to K_Ca_2.1 being localized intracellularly (in analogy to mitochondrial K_Ca_2.2 and K_Ca_2.3 channels [48,49]). However, the channel is likely not active in intracellular membranes such as mitochondria in EwS cells, because the membrane-permeable inhibitor NS8593 fails to affect cell migration (Appendix A). Similar results for non-functional K_Ca_ channels have already been described [50]. The Sontheimer group detected expression of all K_Ca_ channel genes in glioma cells, but they could only measure currents of the K_Ca_1.1 channel across the cell membrane. Neither K_Ca_2- nor K_Ca_3.1-carried membrane currents could be measured. Apparently, cancer cells functionally express only a very limited array of the potentially available ion channels. 

This raises the question of whether the *KCNN1* mRNA is translated into K_Ca_2.1 protein in EwS tumor cells at all. Naturally, we aimed to assess protein expression using Western blot, but we found every commercially available antibody we tested to be unspecific for K_Ca_2.1 (Appendix A). Ion channel proteins often have non-conductive properties, e.g., behaving as membrane anchors [51]. There is, however, no information on non-conducting properties of the K_Ca_2.1 channel, likely because of the lack of a specific probe. By using siRNA against *KCNN1*, however, we could not observe any difference in cell volume homeostasis and RVD (Figure 7C). This evidence argues against a non-conducting channel function. Taken together, we presume that the *KCNN1* mRNA and the K_Ca_2.1 protein—if translated—is most likely degraded or sequestered. A potential mechanism for this could be post-transcriptional mRNA editing that has already been described for ion channels [52,53,54]. To study the events that happen after translation and to better understand the K_Ca_2.1 channel in general, a novel specific K_Ca_2.1 antibody would need to be designed. Another plausible approach would be to conjugate the bee toxin apamin with a fluorophore, similarly to scorpion toxins for detecting K_V_ channels [55].

We detected multiple transcript variants of *KCNN1* in EwS. Among these, the transcript variant 2, termed *KCNN1_EWS,_* is intriguing, as only untranslated exons are altered compared to the canonical transcript variant. Whether this affects channel expression in EwS cells or in a model system still remains to be investigated in more detail. The truncated ΔS2-S3 isoform of *KCNN1* raises the possibility of a dominant negative effect, as earlier described for *KCNN3* [56]. However, we failed to confirm the dominant negative effect when heterologously expressing *EWSR1-FLI1* together with *KCNN1*, which argues against a possible dominant negative effect. In addition, due to the several transcripts found in EwS, we detected exons preserved in every transcript variant (Exons 7–12) in our RNAish study. Whether there is a spatial pattern in the expression of the different transcript variants or whether any specific transcript variant correlates with the clinical features of EwS is an open question to be pursued in future studies. 

A surprising finding was the generally low K^+^ conductivity of the EwS cell membrane. Biophysically, this would make EwS cells ideal models for K^+^ channel transfection. From a physiological point of view, this feature leaves them vulnerable to sudden changes in environmental osmolarity. The question is whether the vulnerability of EwS cells to osmotic stress can also be exploited therapeutically, for example in the form of lavage with distilled water during surgery to lyse any tumor cells remaining after resection. Such a form of tumor therapy is already established, for example, in ruptured hepatocellular carcinoma, where peritoneal lavage with distilled water significantly improves disease-free survival as well as overall survival [57,58]. However, in EwS, it is unlikely that local control and prognosis can be further improved by hypotonic lavage of the surgical site, since tumors located in the extremities can be surgically removed by wide resection—a key component of standard therapy. Moreover, since EwS, unlike hepatocellular carcinoma, is radiosensitive, unresectable local disease can mostly be controlled by radiotherapy. Prognosis limiting in EwS is rather the presence of hematogenous (micro-)metastases, causing distant relapses even after effective local tumor therapy and not easily amenable to osmotic manipulation. Therefore, it would be vital to design a systemic therapy that induces osmotic stress in vivo by selectively targeting tumor cells.

## 5. Conclusions

In summary, numerous Ca^2+^ (regulated) ion channel genes are expressed in EwS that may protect cells from the elevated environmental [Ca^2+^] upon tumor-induced osteolysis. From these genes, we followed up on the prominent expression of *KCNN1,* a channel that is normally only expressed in excitable cells. We elucidated in detail that the oncoprotein EWSR1-FLI1 directly activates an enhancer in the *KCNN1* locus by directly binding to GGAA microsatellites. The activation of the enhancer leads to the transcription of a novel transcript variant *KCNN1_EWS_*. In a large EwS patient cohort, we found that *KCNN1* mRNA does not correlate with any of the investigated clinical parameters. In hindsight, this is not surprising, as we found that K_Ca_2.1, the channel encoded by *KCNN1,* elicits no K^+^ currents in EwS cells and has no measurable implication in numerous aspects of EwS cell physiology. Lastly, we found that the generally low K^+^ conductance of EwS cells leads to an ineffective volume homeostasis, which we could then exploit by inducing EwS cell death by hypoosmotic stress. 

## Figures and Tables

**Figure 1 cancers-14-04819-f001:**
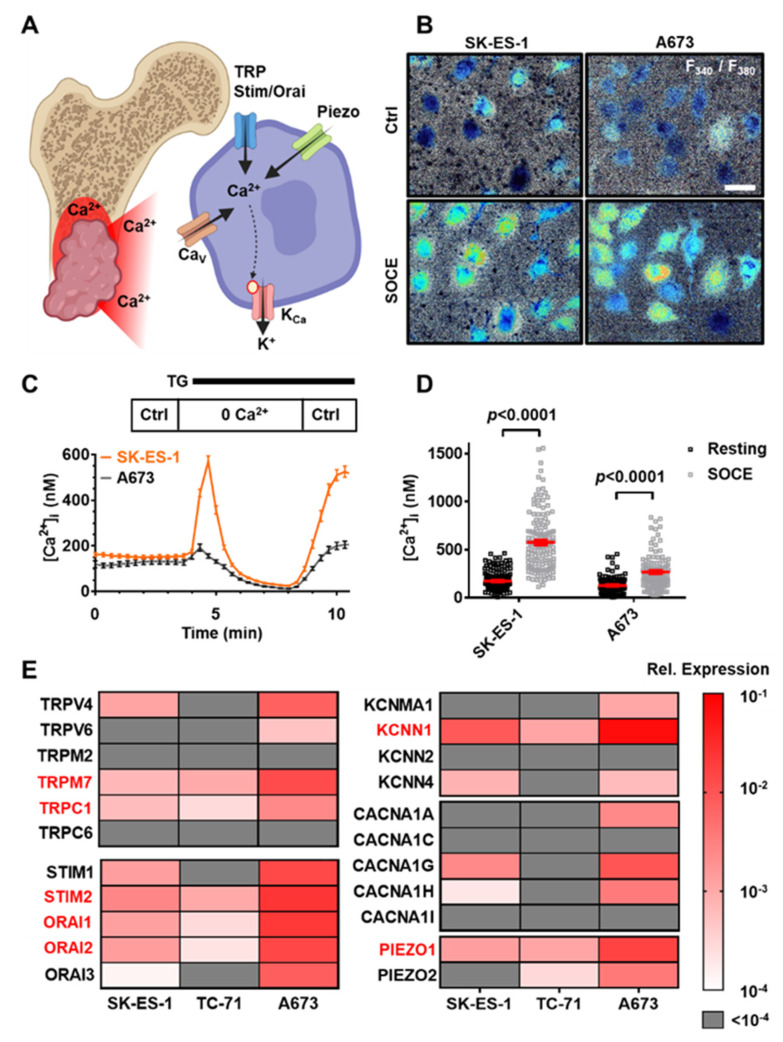
Ewing sarcoma cells express multiple Ca^2+^ (regulated) ion channels. (**A**) Representation of osteolysis-induced Ca^2+^ release from the bone tissue in EwS. Cells can respond to this phenomenon by utilizing Ca^2+^-permeable and Ca^2+^-regulated ion channels prominently from the TRP, Stim, Orai, Piezo, Ca_V_ (encoded by *CACNA* genes), as well as K_Ca_ channels (encoded by *KCNN* genes). (**B**) Representative F_340_/F_380_ ratio images derived from intracellular Ca^2+^ measurements under control conditions (Ctrl) and during store-operated Ca^2+^ entry (SOCE). Brighter and warmer colors indicate higher [Ca^2+^]_i_. Scale bar= 20 µm. (**C**) Summarized Ca^2+^ measurement results of SK-ES-1 (orange) and A673 (black) EwS cells, where individual data points depict the mean [Ca^2+^]_i_ of respective cells at a given time. Resting [Ca^2+^]_I_ was derived from the initial (Ctrl) superfusion; store release was elicited by thapsigargin treatment in the absence of [Ca^2+^]_e_; SOCE can be observed when applying thapsigargin with Ca^2+^-containing Ctrl solution. (**D**) [Ca^2+^]_i_ at resting condition (black) and upon triggering SOCE (peak concentration; grey). Data points depict the [Ca^2+^]_i_ of individual cells. N_SK-ES-1_ = 4, n_SK-ES-1_ = 108; N_A673_ = 4, n_A673_ = 111. (**E**) Heatmap shows ion channel gene expression in EwS cell lines compared to housekeeper genes *GAPDH* and *YWHAZ*. Color code is represented on the right. N = 3, n = 6. Data in (**C**,**D**) are represented as mean ± SEM. (**A**) was created with BioRender.com, accessed on 21 April 2022.

**Figure 2 cancers-14-04819-f002:**
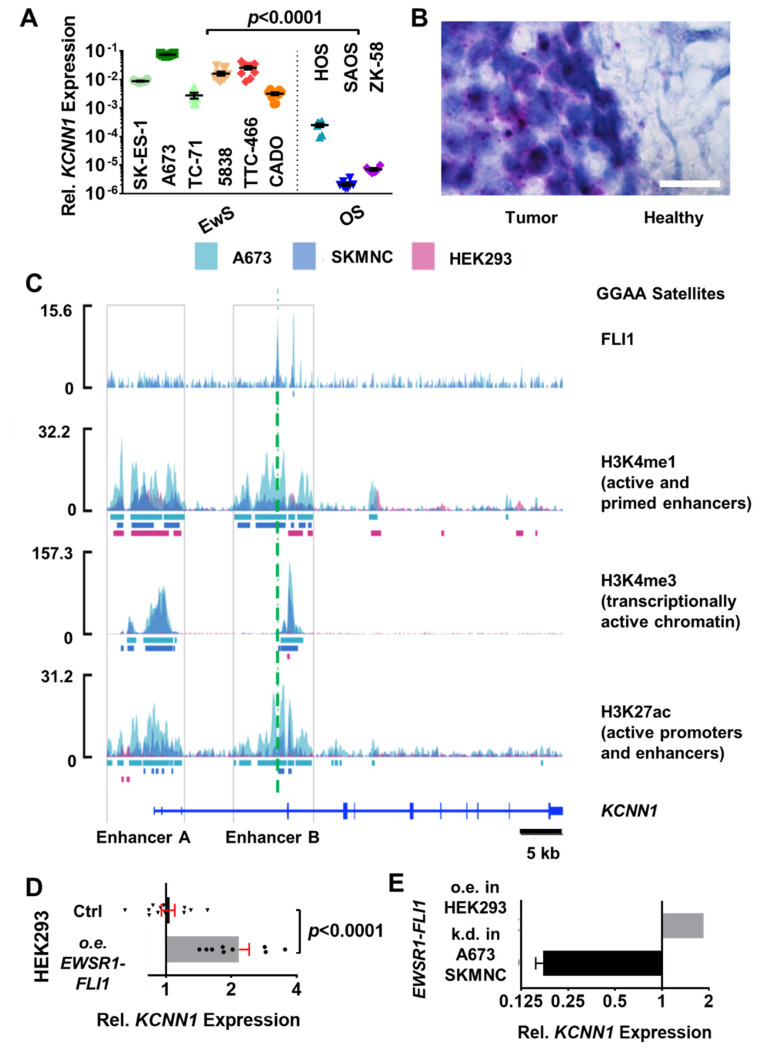
EWSR1-FLI1 drives *KCNN1* expression in Ewing sarcoma. (**A**) Relative expression of *KCNN1* compared to housekeeping genes *GAPDH* and *YWHAZ*. Dashed line separates EwS cell lines (**left**) from osteosarcoma (OS) cell lines (**right**) (N = 3, n = 6 per cell line). (**B**) Representative RNAish brightfield image from an EwS tissue section in a tumor region neighboring healthy bone stroma. White arrows pointing to purple dots indicate *KCNN1* RNAish signals, which are absent in the healthy bone. N = 3, n = 3. (**C**) In A673 and SKMNC Ewing sarcoma cells, ChIP-seq detects strong binding of FLI1 at a GGAA microsatellite (dashed line) in the third intron of *KCNN1*. H3K4me1 marks two poised enhancers at the *KCNN1* locus. Both enhancers are inactive in HEK293 cells, while they are strongly activated in Ewing sarcoma cells, shown by increased H3K27ac. Moreover, around the GGAA microsatellite (green dashed line), the enhancer signal becomes much wider (enhancer B). This leads to transcriptional activation of the gene (H3K4me3). Signal tracks are fold-enhancement relative to input; bars show peaks detected by MACS2 in the respective cell line. (**D**) Relative *KCNN1* expression of HEK293 cells transfected with ctrl vs. EWSR1-FLI1. N = 3, n = 9. (**E**) Knockdown (k.d.) of *EWS-FLI1* in Ewing sarcoma cell lines A673 or SKMNC nearly abolishes *KCNN1* expression (data from [33,35]), while *EWS-FLI1* overexpression (o.e.) in HEK293 cells leads to upregulated *KCNN1* expression compared to control cells (data from [34]). Data in (**A**,**D**,**E**) are represented as mean ± SEM.

**Figure 3 cancers-14-04819-f003:**
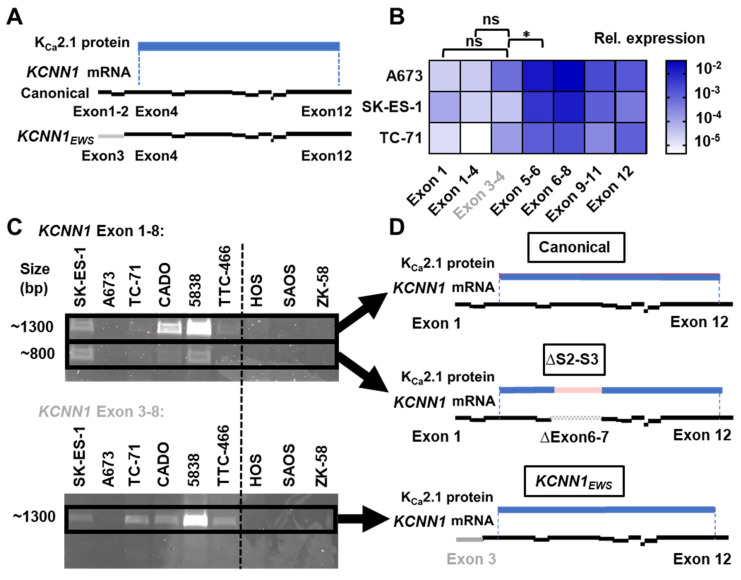
Detection of the *KCNN1_EWS_* transcript. (**A**) Schematic visualization of *KCNN1* transcription and translation. Exon 1–2 (black) and 4–12 correspond to the canonical transcript (NM_002248), whereas alternative transcript (NM_0011386975) expresses exon 3 (grey)-12. Dashed lines represent the start and the end of the open reading frame, and blue line represents the resulting K_Ca_2.1 channel protein. (**B**) Heatmap shows quantification of *KCNN1* exon expression in EwS cell lines A673, SK-ES-1 and TC-71, with color code on the right. N = 3 different EWS cell line passages with a total of n = 14 replicates per data point. * *p* < 0.05. (**C**) Representative polyacrylamide gels derived from RT-PCR of the canonical transcript (top, exon 1–8) and the alternative transcript (bottom, exon 3–8). Dashed line separates EwS cell lines (**left**) from OS cell lines (**right**). N = 3 different EWS cell line passages with n = 3 technical replicates. (**D**) Illustration—explained in (**A**)—shows *KCNN1* mRNA sequences derived by capillary sequencing from transcripts highlighted in (**C**) with bold rectangles. Full PCR gel images can be found at Appendix A.

**Figure 4 cancers-14-04819-f004:**
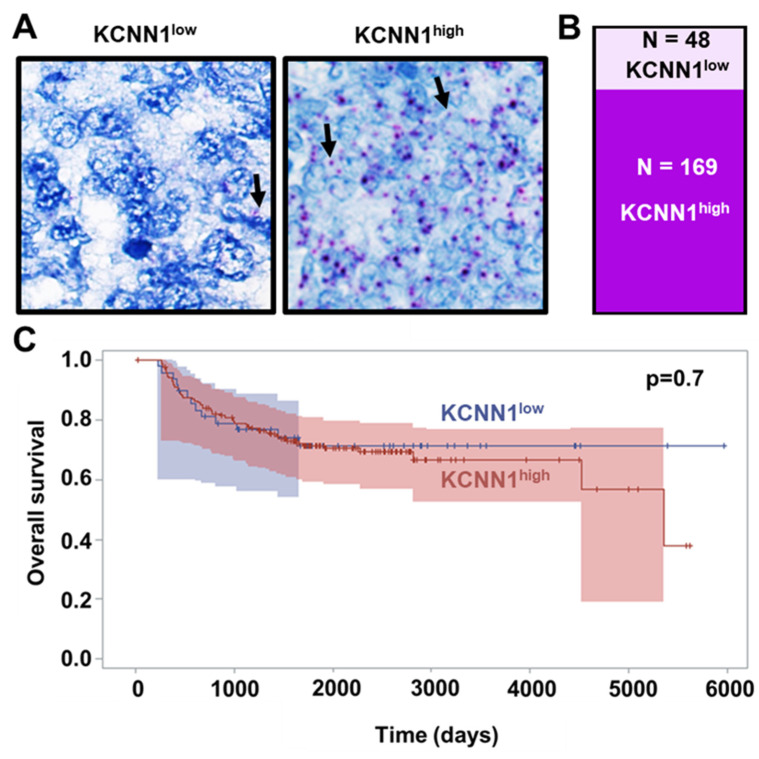
*KCNN1* expression does not correlate with overall survival of Ewing sarcoma patients. (**A**) Representative RNAish brightfield image from EwS tissue microarrays. Black arrows pointing at purple dots indicate *KCNN1* RNAish signal. For further analysis, we differentiated between *KCNN1*^low^ (left) and *KCNN1*^high^ (right) TMA samples. (**B**) Slice plot shows the relative amount of *KCNN1*^low^ *KCNN1*^high^ TMA sections in the cohorts (**C**) Kaplan–Meyer curve indicating the overall survival of *KCNN1*^low^ (blue) versus *KCNN1*^high^ (red) EwS patients in respect to the time after diagnosis. Survival functions between the groups were analyzed by a log-rank test. Fischer’s exact test was used to calculate *p*-values Shaded areas indicate 95% CI.

**Figure 5 cancers-14-04819-f005:**
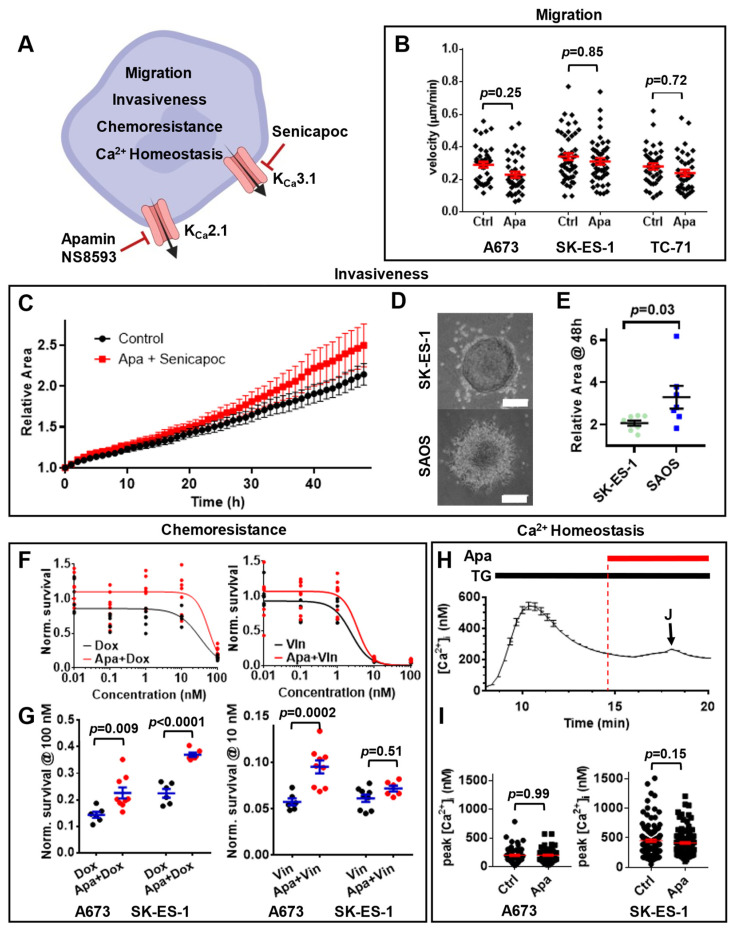
K_Ca_2.1 is non-functional in Ewing sarcoma cells. (**A**) Illustration of investigated cell physiological processes assessed for K_Ca_ channel dependence. Red arrows indicate pharmacological inhibitors of K_Ca_2.1 and K_Ca_3.1, respectively. (**B**) Cell migration velocities of EwS cells in each condition were derived from individual trajectories. Each dot represents the mean velocity of one cell. (**C**) Increase in the relative area of SK-ES-1 spheroids within 48 h. Spheroids were monitored under control conditions or treated with 100 nM apamin and 10 µM senicapoc N = 4, n = 8. (**D**) Representative images of matrix-embedded SK-ES-1 EwS cell spheroids and SAOS osteosarcoma cell spheroids acquired after 48 h. Scale bar = 200 µm. (**E**) Scatter plot indicates relative area of SK-ES-1 and SAOS spheroids. N = 4, n_SK-ES-1_ = 8, n_SAOS_ = 7. (**F**) Survival after 72 h is depicted as a function of increasing concentration of doxorubicin (Dox, left) and vincristine (Vin, right), respectively. Data are derived from MTT assays. They were normalized to vehicle-treated cells (control). Additional treatment with 100 nM apamin is indicated in red. Data points were fitted with four-parameter dose–response curves (solid lines). (**G**) Scatter plots depict normalized cell survival at 100 nM Dox (**left**) and 10 nM Vin (**right**). EwS cell survival was calculated and normalized as detailed in (**F**). N = 3, n = 6. (**H**) Summary of intracellular Ca^2+^ measurements of SK-ES-1 cells after eliciting SOCE with thapsigargin (TG). At t = 15 min (red dashed line), cells were additionally superfused with 100 nM apamin (Apa, red line). Data in (**I**) are derived at the time point indicated by J (arrow). Each data point depicts the mean of N = 4, n = 137 cells. (**I**) Scatter plot compares peak [Ca^2+^]_i_ under control with apamin superfusion at time point indicated in (**H**). N_A673_ = 3, n_A673_ = 62, N_SK-ES-1_ = 4, n_SK-ES-1_ = 137. Data are shown as mean ± SEM. (**A**) was created with BioRender.com, accessed on 21 April 2022.

**Figure 6 cancers-14-04819-f006:**
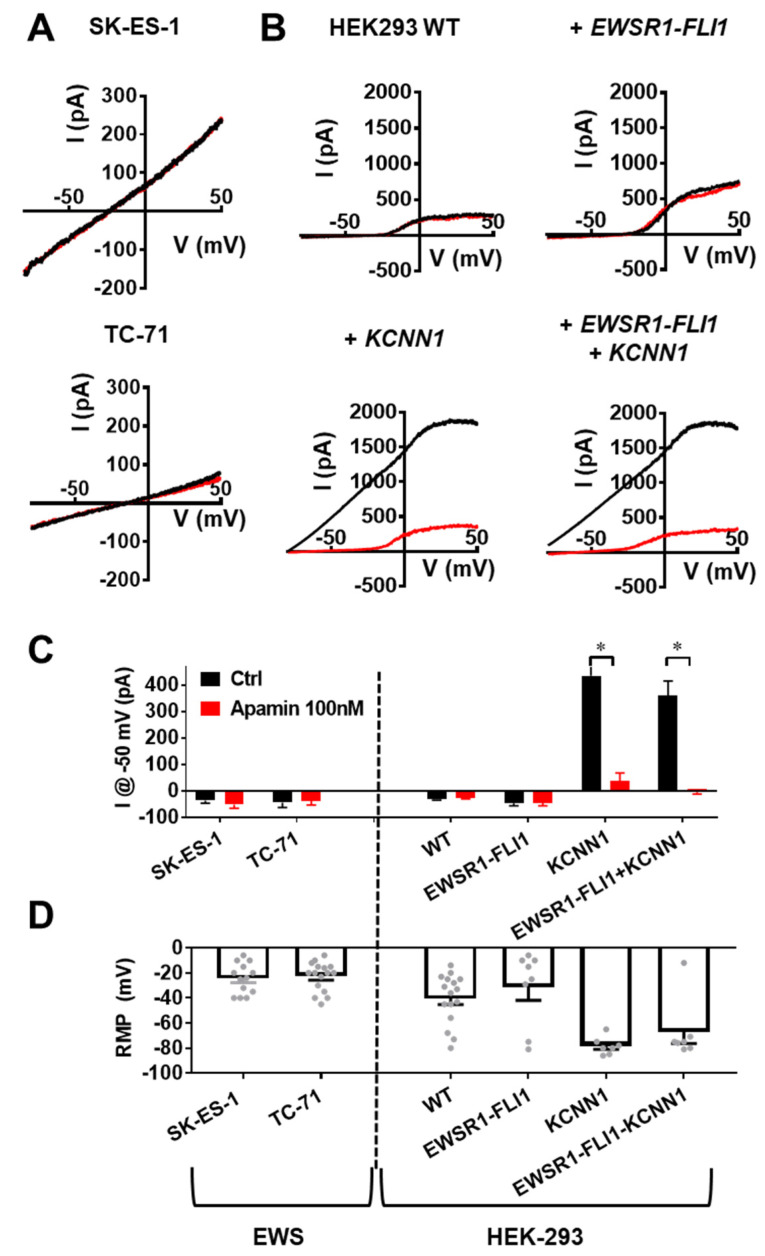
K_Ca_2.1 channels are non-conducting in EwS cells. (**A**) Representative whole-cell patch-clamp recordings of SK-ES-1 (top) and TC-71 (bottom) cells under control conditions (black) or in the presence of 100 nM apamin (red). Voltage ramp was applied between −80 and +50 mV in 5 s. (**B**) Representative whole-cell patch-clamp recordings of HEK293 cells transfected with either control (HEK−293 WT, top left), *EWSR1-FLI1* (top right), *KCNN1* (bottom left) or both *EWSR1-FL1* and *KCNN1* (bottom right), under control conditions (black) or with 100 nM apamin superfusion (red). (**C**) Quantification of whole-cell currents at −50 mV under control conditions (black) or in the presence of 100 nM apamin (red). (**D**) Resting membrane potential (RMP) of cells measured at the beginning of each experiment. Dashed line separates EwS cell lines (left) from transfected HEK293 cells (right) in (**C**,**D**). Data are represented as mean ± SEM. * *p* < 0.05.

**Figure 7 cancers-14-04819-f007:**
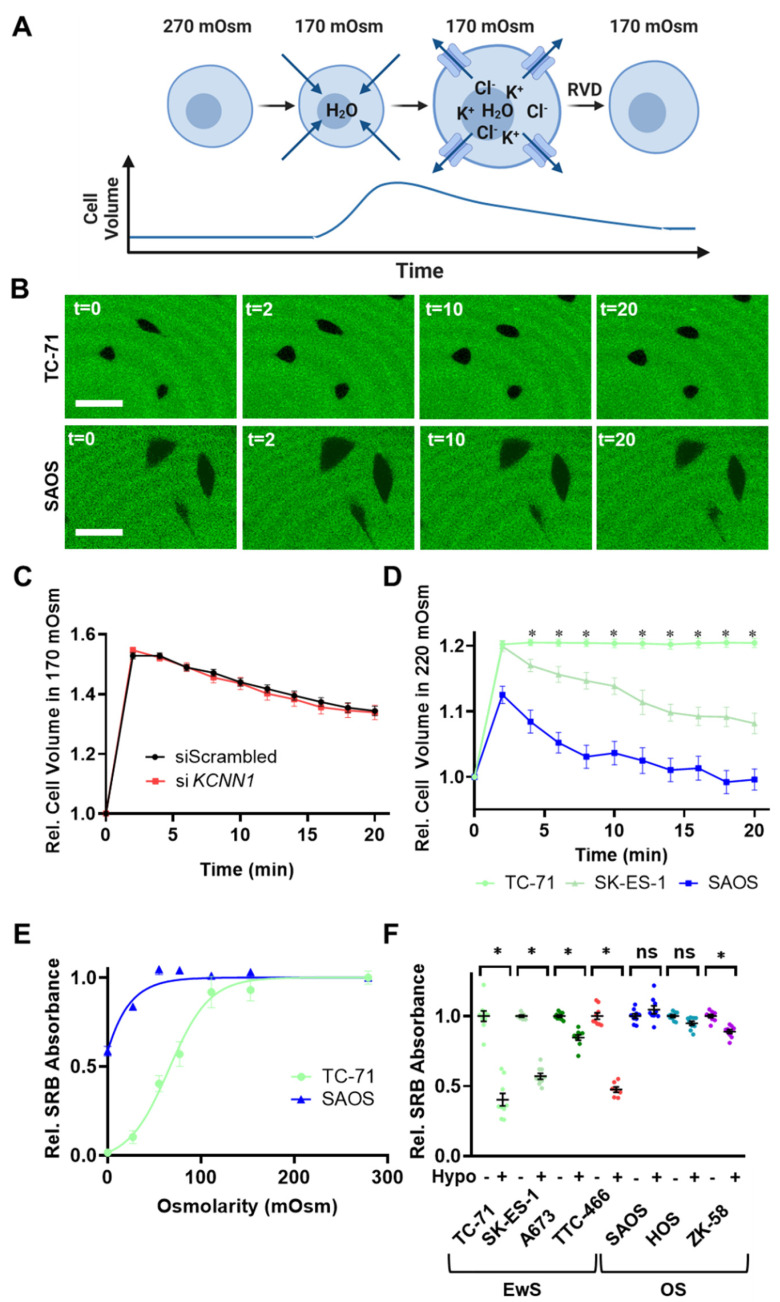
Poor volume regulation of EwS cells results in vulnerability for osmotic stress. (**A**) Schematic figure of the process of regulatory volume decrease (RVD). When a cell is exposed to a hypotonic solution, it swells due to the osmotic gradient. The subsequent RVD is a counter-regulatory mechanism which is mainly ensured by ion-channel-mediated K^+^- and Cl^−^ efflux through the plasma membrane, which is followed by a water loss. (**B**) Representative confocal images of TC-71 EwS (top) SAOS osteosarcoma (bottom) cells (black) which were exposed to a hypotonic solution (220 mOsm) containing high-molecular-weight FITC-dextran (green). Cells first swell, which could be observed at t = 2 min. Thereafter, cells undergo RVD aiming to regenerate their original volume. Scale bar (lines)=50 µm. (**C**) RVD performance of TC-71 cells treated with si*KCNN1* (red, N = 4, n = 32) or with scrambled siRNA (black, N = 4, n = 38), indicated by changes in their relative cell volume in a hypoosmotic solution (170 mOsm). The volume is normalized to their initial volume in an isosmotic solution. (**D**) Comparison of relative cell volume changes of EwS cell lines TC-71 (green, N = 4, n = 26), SK-ES-1 (teal, N = 3, n = 42) and the osteosarcoma cell line SAOS (blue, N = 3, n = 33) after exposure to a hypoosmotic solution (220 mOsm). (**E**) Cell survival of TC-71 EwS cells (green, N = 3, n = 9) and SAOS osteosarcoma cells (blue, N = 3, n = 9). Cells were treated with different hypotonic solutions for 15 min, and cell survival is indicated by the sulforhodamine B (SRB)-absorbance. The SRB absorbance was normalized to isosmotic solution. A sigmoidal dose–response curve was fitted to the viability of both cell lines to determine the IC_50_. (**F**) Cell survival of EwS cell lines TC-71, SK-ES-1, A673 and TTC-466 and osteosarcoma cell lines SAOS, HOS and ZK-58 derived from relative SRB absorbance. Isoosmolar conditions (Hypo -) were compared to 55 mOsm hypoosmotic solutions (Hypo +). * Data are represented as mean ± SEM. *p* < 0.05, ns: not significant. (**A**) was created with BioRender.com, accessed on 21 April 2022.

**Table 1 cancers-14-04819-t001:** Characteristics of cell lines used for the experiments.

Cell Line	Diagnosis	Tissue of Origin	EWSR1-Rearrengement	RRID
SK-ES-1	Ewing Sarcoma	Bone	EWS-FLI1	CVCL_0627
TC-71	Ewing Sarcoma	Bone, Humerus	EWS-FLI1	CVCL_2213
A673	Ewing Sarcoma	Muscle	EWS-FLI1	CVCL_0080
TTC-466	Ewing Sarcoma	Lung metastasis	EWS-ERG	CVCL_A444
SAOS	Osteosarcoma	Bone	none	CVCL_0548
HOS	Osteosarcoma	Bone	none	CVCL_0312
ZK-58	Osteosarcoma	Bone	none	CVCL_9920

## Data Availability

All scripts and analyzed data are available from the corresponding author upon reasonable request.

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
