# Peer review of "Relevance of Abnormal KCNN1 Expression and Osmotic Hypersensitivity in Ewing Sarcoma"

_cancers, 2022, doi:10.3390/cancers14194819_

Round 1

Reviewer 1 Report

Treatment of Ewing sarcoma is still a challenge and several approaches, especially for metastatic disease, have only limited impact. The manuscript gives insight in the overexpression but functional inactivity of KCNN1. Though, disappointingly KCNN1 appears to be without functional relevance the study discovers the interesting fact that EwS cell lines are susceptible to hypoosmotic shock.

Methodically the study is well performed and merits publication since it increases the available knowledge about EwS, a rare tumor mainly affecting children and adolescents.

Minor points

Missing word and spelling

Page 17: After incubation with hypotonic solutions of 77 mOsm or less, TC-71 EwS cell survival is considerably lower than that of SAOS osteosarcoma cells

Page 20: This evidence argues against a non-conducting channel function

Author Response

We thank the Reviewer for reviewing our manuscript. We corrected these errors in our new version. 

Reviewer 2 Report

In this article, the authors explained the relevance of KCNN1 expression and osmatic hypersensitivity in Ewing sarcoma. 

1. 48 hours is a very shorter period for the spheroidal experiment. can the authors use a low number of cells and continue the experiment for a longer period of time ( at least 7 days)? 

2. Can the authors use high resolution image for Figure 2B. 

Author Response

We thank the Reviewer for the constructive feedback and the time and effort spent on the revision.

  1. We conducted our spheroid migration assays for 48 hours due to multiple reasons: A) especially SAOS spheroids were very active and frequently migrated outside the field of view after 48 hours. Therefore, the videos could not be further evaluated afterwards. B) It is not possible to control the environmental variables exactly for 7 days in our system, because we do not open the cell migration chambers nor exchange media. Moreover, when observing cells for a long time without exchanging medium, other problems would arise, e.g. apamin and senicapoc instability would be a concern for a long time. C) We agree that a low cell count would generally help for long-term spheroid observation. We indeed did preliminary experiments determining the optimal cell count for the spheroids, but Ewing sarcoma cell spheroids did not form when we used lower cell numbers. D) In at least our setting, the live cell imaging method is so sensitive and consistent that, given the linear progression as seen on Fig. 5D, an observation period of 48 h is more than sufficient.
  2. We provided a high-resolution version of the image in Figure 2B.

Round 2

Reviewer 2 Report

No further comments